# Autophagy/Mitophagy Regulated by Ubiquitination: A Promising Pathway in Cancer Therapeutics

**DOI:** 10.3390/cancers15041112

**Published:** 2023-02-09

**Authors:** Seung-Cheol Jee, Heesun Cheong

**Affiliations:** Division of Cancer Biology, Research Institute, National Cancer Center, Goyang-si 10408, Republic of Korea

**Keywords:** autophagy, mitophagy, ubiquitination, deubiquitination, cancer

## Abstract

**Simple Summary:**

Autophagy and mitophagy are important processes in the regulation of cancer progression. Although autophagy and mitophagy have dual roles in cancer, targeting their regulation has potential for developing an effective cancer treatment strategy. Thus, it is important to understand how ubiquitination and deubiquitination of autophagy-related proteins are regulated to exploit autophagy and mitophagy during cancer development.

**Abstract:**

Autophagy is essential for organismal development, maintenance of energy homeostasis, and quality control of organelles and proteins. As a selective form of autophagy, mitophagy is necessary for effectively eliminating dysfunctional mitochondria. Both autophagy and mitophagy are linked with tumor progression and inhibition. The regulation of mitophagy and autophagy depend upon tumor type and stage. In tumors, mitophagy has dual roles: it removes damaged mitochondria to maintain healthy mitochondria and energy production, which are necessary for tumor growth. In contrast, mitophagy has been shown to inhibit tumor growth by mitigating excessive ROS production, thus preventing mutation and chromosomal instability. Ubiquitination and deubiquitination are important modifications that regulate autophagy. Multiple E3 ubiquitin ligases and DUBs modulate the activity of the autophagy and mitophagy machinery, thereby influencing cancer progression. In this review, we summarize the mechanistic association between cancer development and autophagy/mitophagy activities regulated by the ubiquitin modification of autophagic proteins. In addition, we discuss the function of multiple proteins involved in autophagy/mitophagy in tumors that may represent potential therapeutic targets.

## 1. Introduction

The ubiquitin–proteasome system (UPS) and autophagy are important for the maintenance of cellular homeostasis. A variety of stimuli such as oxidative stress, heat stress, and DNA damage induces the accumulation of abnormal proteins in the ER [1]. Moreover, in normal cells, approximately 30% of newly synthesized proteins are misfolded [2]. These abnormal proteins are eliminated through UPS and autophagy, which contributes to the maintenance of cellular proteostasis.

Ubiquitination is important for the degradation of proteins, thereby regulating their function. The inhibition of ubiquitination is responsible for the development of certain diseases and cancers [3]. In particular, the regulation of metabolic signaling pathways and transcription factors, such as oncogenes or tumor suppressors, is closely associated with cancer metabolism through ubiquitination [4]. Deubiquitination maintains homeostasis, and deubiquitinating enzymes (DUBs) regulate many cellular processes. Studies have shown that the activation of DUBs is also dysregulated in cancer [5].

Recent reports have suggested that UPS is interconnected with autophagy, and many studies have focused on the relationship between autophagy and the ubiquitination process. Autophagy is known for its pathophysiological role in cancer and is involved in the elimination of damaged proteins, pathogens, and aggregates, which influences the cell death pathway [6,7,8]. Moreover, regulation of UPS and/or autophagy has effects on one another [9]. One predominant hypothesis is that UPS inactivates autophagy-mediated lysosomal degradation and vice versa; however, other theories suggest that UPS and autophagy complement each other through close molecular interactions between these two protein degradation pathways.

These studies indicate that understanding how autophagy proteins are regulated by ubiquitination and deubiquitination and the contribution of UPS-mediated degradation are important for understanding cancer progression. Here, we summarize the role of ubiquitination and deubiquitination in autophagy and mitophagy and further discuss how their regulation influences cancer development. In the current review, we highlight the role of mitophagy and autophagy as cancer regulators through the ubiquitination and deubiquitination system in addition to their potential as therapeutic targets. 

## 2. Ubiquitination

Ubiquitin, which consists of 76 amino acids, is conjugated with substrate proteins. The degradation of ubiquitin-coupled proteins is a part of many biological processes, such as cell cycle progression, apoptosis, signal transduction, and transcriptional regulation [10,11]. This process is initiated through the formation of a thioester bond between cysteine at the active site of ubiquitin-activating enzymes (E1s) and the carboxyl end glycine residue of ubiquitin, which activates ubiquitin, and the activated ubiquitin is transferred to the cysteine residue of ubiquitin-conjugating enzymes (E2s). Finally, ubiquitin ligases (E3s) transfer ubiquitin from the E2 to a lysine residue on the substrate by forming an isopeptide bond. Ubiquitin is attached through at least seven lysine residues or N-terminal Met, which regulates protein fate based on the type of polyubiquitination. Previous studies have demonstrated that K6-linked ubiquitination is involved in DNA damage repair [12]. K11-linked ubiquitination is related to both proteolysis and non-proteolytic functions. K27-linked ubiquitination regulates mitochondrial autophagy, and K29-linked ubiquitination is associated with the degradation of proteins associated with ubiquitin-fusion degradation (UFD) [13,14]. K48-linked ubiquitination is associated with proteasomal recognition and degradation. K63-linked ubiquitination regulates signaling assemblies and is also involved in the lysosomal degradation of target proteins [15,16,17].

Ubiquitinated proteins are degraded through the ubiquitin proteasomal pathway (UPP). Many studies have indicated that UPP is associated with many diseases and cancer [18]. Typically, the regulation of UPP determines the proteasomal levels of E3 ligases. Evidence from one study supports the notion that more efficient treatment of diseases and cancer regulation may be achieved by targeting E3 ubiquitin ligases rather than E1 enzymes [19]. In humans, approximately 1000 E3 ligases have been identified, and E3 ligases possess specificity against their substrates, which is important to the whole process of ubiquitination.

Deubiquitination facilitates the maintenance of homeostasis by removing ubiquitin from ubiquitinated proteins, which is catalyzed by DUBs. Multiple studies have demonstrated that DUBs are involved in the maintenance of protein stability, editing of ubiquitin chains, and processing of ubiquitin precursors, which are associated with disease development and progression [20,21]. 

Many studies have indicated that DUBs are involved in tumorigenesis. DUBs play dual roles as oncogenes or tumor suppressors [21,22]. Certain DUBs modulate the epigenetic effect in cancer cells, which induces tumorigenesis, whereas DUBs also directly regulate oncoproteins or tumor suppressor proteins [23,24]. Finally, DUBs attenuate cancer-related processes by stabilizing participant proteins for respective cellular processes. Among the various cancer-related cellular processes, autophagy is a promising target pathway for drug development. Accordingly, it is important to understand how ubiquitination and deubiquitination regulate autophagy in cancer.

## 3. Autophagy Machinery

Autophagy is a highly conserved degradation system that is categorized into three major types based on the following morphological features: macroautophagy, microautophagy, and chaperon-mediated autophagy (CMA) [25]. Autophagy is normally active at a basal level and is upregulated following cellular stress and stimuli, such as hypoxia, oxidative stress, nutrient starvation, pathogen invasion, and mTOR signaling pathway inhibition [26,27]. Most autophagy-inducing signals converge at the level of mTOR complexes, which coordinate catabolic and anabolic processes. mTOR complex 1 (mTORC1) is a well-known upstream regulator of autophagy that is sensitive to nutrient conditions [28]. When these signals are inactivated by low levels of nutrients or growth factors, the development of a phagophore is initiated through double-membranes that are linked with LC3-II. The phagophore then engulfs the organelles and cytosolic components to be degraded to form an autophagosome, which then fuses with lysosomes and late endosomes to form an autolysosome with an acidic interior and hydrolytic environment that facilitate the degradation of its contents. Hence, autophagic cargo-derived materials such as amino acids, fatty acids, and nucleotides from damaged organelles together with unnecessary cytosolic components and misfolded proteins are degraded and recycled for reuse in biosynthetic processes and cellular bioenergetics [25]. Autophagosome biogenesis is organized by autophagy-related gene (ATG) proteins. At least 40 ATG proteins have been identified in yeast and mammals [29]. ATG protein function is classified based on physical interaction and their role in nonselective and selective autophagic function (Table 1). Specifically, a core molecular machinery has an important role in forming sequestered vesicles [30]. The ULK complex, a mammalian ATG1 homolog that contains ATG13, ATG101, and FIP200, is involved in the termination of growth-related signaling cascades and the initiation of autophagy [31]. ULK1 is a downstream substrate of mTORC1. Under starvation conditions, mTORC1 activity decreases, which results in the activation of the ULK complex through site-specific phosphorylation of multiple components [32]. In addition, AMPK regulates ULK1 activity via phosphorylation, which is associated with mTORC1 activity [33]. The ULK complex induces the production of class III phosphatidylinositol 3-kinase (PI3KC3) complex components, including ATG14, BECN1, p150, PIK3R4, and Vps34 (PI3KC3). This results in the production of phosphatidylinositol 3-phosphate (PI3P) at the phagophore site, which recruits ATG proteins. Subsequently, the autophagosome membrane is elongated, which is dependent upon two ubiquitin-like conjugation pathways. First, ATG5, and ATG12 are conjugated, which requires ATG7 and ATG10. ATG5 conjugated with ATG12 interacts noncovalently with ATG16L1 to form an oligomeric complex. Then, ATG4b hydrolyzes pro-LC3-like ATG8 (several homologs form; MAP1LC3 and GABARAP) to release the C-terminal glycine for conversion into LC3-II, which becomes LC3-II once combined with phosphatidyl ethanolamine (PE) through catalysis of the ATG7 (E1-like), ATG3 (E2-like), and ATG16L1-ATG5-ATG12 (E3-like) complexes [34].

WIPIs are represented by four subclasses and are involved in recruiting ATG12-5-16L1 and LC3 lipidation [35]. Specifically, WIPI1, and WIPI2 are associated with ATG12-5-16L1 recruitment and conjugation of LC3 with the synthetic membrane. Additionally, WIPI2 has been identified as a CUL4-RING ubiquitin E3 ligases (CRL4s) substrate, which leads to proteasomal degradation and polyubiquitination of WIPI2, thereby regulating autophagy [36]. WIPI3 and WIPI4 support the recruitment of lipids to nascent phagophores by formation of the ATG2 complex, which, through its interaction with PI3P, is required for autophagic membrane formation, whereas the WIPI–ATG2 complex is essential for the initiation of autophagy [37]. ATG9, a unique protein of the transmembrane core ATG, is linked to the elongation and nucleation process through interactions with ATG2 and WIPIs. The regulation of autophagosome formation only occurs when ATG11 recruits ATG9 to prApe1 via the selective autophagy-like cytoplasm-to-vacuole targeting pathway [38]. 

Lipidated LC3-II attaches to both sides of the phagophore membrane but eventually fuses to lysosomal integral membrane proteins, such as RAB and SNARE, with the outer membrane of the autophagosome, resulting in autophagosome fusion with late endosomes or lysosomes. Recent studies have suggested that the SNARE complex is important for autophagosome–lysosome fusion. There are four types of SNARE-mediated fusion: Q_a_, Q_b_, Q_c_, and R of SNARE motifs [39]. STX17 and SNAP29, a major autophagosomal Q-SNARE protein, are involved in autophagosome maturation. STX17 and SNAP29 form the autophagosome Q_abc_ bundle, which subsequently forms a complex with a lysosome-localized R-SNARE, such as VAMP7, VAMP8, and YKT6, during autophagosome–lysosome fusion [40].

Understanding the processes of molecular regulation occurring at each step of autophagy will provide insight for effective targeting of autophagy given its involvement in the development of diseases such as cancer.

**Table 1 cancers-15-01112-t001:** Specific roles of ATG proteins in autophagy.

	Gene (Mammals)	Protein Functions	References
ULK kinase complex	ULK1/2	Regulation of mTOR and autophagy, initiation of autophagy, and termination of several signaling cascades	[41,42]
	ATG13	Acts as an adaptor to form ULK complexes by recruiting ULK1, ATG101, and FIP200 and is involved in autophagy initiation	[43]
	FIP200	Autophagy initiation, autophagosome biogenesis, and interacts with autophagy receptors	[44,45]
	ATG101	Stability and phosphorylation of ATG13 and ULK1	[46]
Class III PI3K complex	Beclin 1	Involved in pro-survival autophagy or pro-apoptotic responses	[47,48,49]
	ATG14	Involved in the assembly of the specific autophagic complex and fusion of autophagosomes to endolysosomes	[50,51]
ATG12-conjugation complex	ATG5	Involved in LC3 lipidation and autophagosome formation	[52,53]
	ATG12	Involved in phagophore elongation, a ubiquitin-like modifier that forms the ATG12–ATG5 complex, which acts as an E3 complex in ATG8–PE conjugation	[54,55]
	ATG7	Acts as an E1-like enzyme for ubiquitin-like proteins, such as ATG8 and ATG12, and regulates autophagosome assembly	[56,57]
	ATG10	Act as an E2-like enzyme in catalyzing the conjugation of ATG12 to ATG5 and is involved in autophagosome formation.	[58]
LC3-conjugation complex	ATG3	Acts as an E2 ubiquitin-like enzyme with ATG8 and ATG12 and is involved in phagophore elongation and LC3 lipidation	[59,60]
	ATG16L1/2	Involved in autophagosome formation	[61]
	ATG4	Acts as a deubiquitinating enzyme and is involved in LC3/GABARAP processing	[62,63]
	LC3, GABARAPs	Involved in autophagosome biogenesis and maturation	[64]
ATG9/ATG2/ATG18 trafficking system	WIPIs	Involved in autophagosome, autophagy initiation, and ATG8 lipidation	[35,37]
	ATG2	Involved in formation of the autophagosome	[65,66]
	ATG9	Interacts with the ATG2/WIPI complex and acts as lipid mobilization during autophagosome progression	[67,68]

## 4. Role of Autophagy in Cancer

### 4.1. The Relationship between Autophagy and Cancer

Autophagy is vital in organismal development, assisting in the regulation of the adaptive immune system, the maintenance of energy homeostasis, and quality control of organelles and proteins [69,70]. Accordingly, autophagy is considered an important biological process and is closely linked to various diseases. Moreover, autophagy plays a complex role in cancer, which is not fully understood [71].

Autophagy exhibits both tumor-suppressing and tumor-promoting functions depending on the cellular context of tumors, such as cancer stage, primary type, and the tumor microenvironment. The functional complexities of autophagy have been reported because of its association with multiple oncogenes or tumor suppressors. The tumor-suppressing role of autophagy was initially identified in a study of the BECN1 gene, which is monoallelically deleted in dominant portions of human breast, ovarian, and prostate tumors [72]. Homozygous BECN1 knockout mice are embryonically lethal, whereas heterozygous BECN1 expression in mice results in spontaneous tumorigenesis. Mice with heterozygotic BECN1 develop lymphoma, hepatocellular carcinoma, and lung adenocarcinomas [73]. 

Deletion of ATG5 results in a tumor-suppressive phenotype, similar to abnormal BECN1 in cancer models, but only benign hepatomas in a mouse liver model [74]. ATG2B and ATG5 suppress cancer stemness by inducing autophagy in TNBC [75]. These results suggest that deficient autophagy enhances tumor initiation at the early stages of liver cancer but does not contribute to further progression of malignant tumors. In colorectal and gastric cancer patients, a reduction in autophagy because of a mutation in ATG2, ATG5, ATG9, and ATG12 was observed [76]. Moreover, ATG12 inhibits anti-apoptotic BCL-2 expression, which inhibits tumor cell survival [77]. These results indicate that autophagy has a tumor-suppressive role in cancer.

The autophagy substrate protein p62/SQSM1 is positively associated with tumor progression. Studies have shown that abnormal p62 accumulation is associated with various cancers [78,79,80,81,82]. Autophagy deficient cells accumulate p62 and induce metabolic stress, which results in oxidative stress and damaged mitochondria [83]. When exposed to metabolic stress, autophagy contributes to survival. Other studies have provided evidence that autophagy induces tumor cell survival and progression [84,85]. Moreover, autophagy-related genes, such as LC3, GABARAP, and ATG8, are induced in tumors compared with normal tissues and are associated with a poor diagnosis [86]. One study suggested that EI24 (etoposide-induced gene 2.4 kb; PIG8, p53-induced gene 8) is a key component of autophagy and promotes the proliferation of pancreatic tumor cells [87].

Autophagy has a dual role in cancer depending on tumor stage and type [88]. Autophagy can regulate genome stability and prevent cell damage at early stages of cancer progression and inhibit the accumulation of p62 aggregates, thereby preventing tumor progression [89]. However, at later stages, autophagy acts as a defense mechanism, maintains mitochondrial function, and attenuates DNA damage, resulting in cancer cell survival through resistance to various stimuli such as nutrient deprivation, DNA damage, hypoxia, and chemotherapy [90]. Multiple studies using genetically engineered mouse models of cancer have suggested support for the cancer-promoting roles of autophagy, despite the fact that induction of tumor initiation by autophagy depletion and genetic deletion of ATG5 and ATG7 in mice ultimately decreases malignant tumor progression in tissue-specific tumor models that are spontaneously derived from known oncogenes [91,92,93]. Consequently, autophagy is interconnected with tumor metabolic alteration and supports tumor progression [94,95]. These results indicate that autophagy is clearly involved in cancer progression; however, its roles in different cancers varu depending on the cancer type, stage, and tumor microenvironment (TME).

### 4.2. Machinery for Ubiquitination and Deubiquitination of Autophagy in Cancer

Recently, numerous reports have suggested a role for ubiquitination/deubiquitination in modulating autophagy [96,97]. In this part of the review, we discuss how certain autophagy-related proteins are regulated by various molecular modes of ubiquitin modification [98,99]. As previously discussed, ubiquitination, and deubiquitination are important for cell function and homeostasis. The modulating systems of ubiquitin are important for proteostasis in cells, which is regulated by autophagy and UPS. Autophagy is regulated by ubiquitination through at least two pathways. First, the stability of upstream autophagy regulators and autophagy machinery are controlled through ubiquitination. Second, protein proximity, or interactions between autophagy components and ubiquitinated proteins are facilitated by the recruitment of autophagy adaptors, such as histone deacetylase 6 (HDAC6), ubiquitin receptor nuclear dot protein 52kd (NDP52), and p62, thereby promoting autolysosome formation [100].

As a key factor in autophagy initiation, ULK1 levels are regulated through ubiquitination and deubiquitination by E3 ubiquitin ligases and DUBs. TRAF6 induces polyubiquitination of ULK1 via Lys63-linked ubiquitin and promotes autophagy [101]. TRAF6–ULK1-dependent activation of autophagy plays a role in chronic myeloid leukemia drug resistance [102]. Conversely, TMEM189 (transmembrane protein 189) disrupts the interaction between TRAF6 and ULK1 and inhibits K63-linked polyubiquitination of ULK1, resulting in reduction of autophagy and induction of tumorigenesis [103]. Moreover, we and another group reported that NEDD4L, an E3 ubiquitin ligase, interacts with ULK1, which results in its ubiquitination, with a subsequent reduction in ULK1 levels and autophagy [104,105]. These phenotypes attenuate cancer cell survival and in vivo tumorigenesis in pancreatic cancer model [104]. Modulation of DUBs affects ULK1 ubiquitination, thereby regulating ULK1 kinase activity and protein stability [106]. Ubiquitin-specific protease 20 (USP20) induces deubiquitination of ULK1, which maintains basal ULK1 levels by preventing ULK1 degradation, and it directly affects autophagy initiation [107]. Finally, USP20 interacts with ULK1 and suppresses autophagy termination under starvation conditions. These studies demonstrate that the balance of ubiquitination and deubiquitination of ULK1 is important for regulating the process of autophagy, and modulating ULK1 stability is a potential strategy for the treatment of cancer.

The class III PI3K (PI3KC3) complex includes four proteins, ATG14L, Beclin-1 (BECN1), BECN-1 regulated autophagy protein 1 (AMBRA1), and VPS34, which are required to initiate phagophore nucleation. Several studies have shown that ubiquitination and deubiquitination of PI3KC3 complex regulate autophagy activity. As E3-ubiquitin ligases, CUL3, NEDD4, TRAF6, TRIM59, and RNF216 ubiquitinate Beclin-1, an autophagy-related protein [108,109,110]. In particular, a recent study demonstrated that CUL3 inhibits autophagy by ubiquitinating and degrading Beclin-1, thus promoting tumor progression [111]. In contrast, deubiquitination of BECN1 is induced by several USPs. For example, USP14 regulates autophagy by suppressing the K63 ubiquitination of Beclin-1 [112]. The DUB Atax3 removes Lys48-linked ubiquitin chains from Beclin-1, which increases the stability of the target in a Parkinson’s disease model [113]. This protein is also regulated by USP19-mediated hydrolysis of Lys11-linked ubiquitin for maintaining Beclin-1 levels to induce autophagic activity and modulate antiviral immunity [114]. In cancer models, USP8, and USP15 negatively regulate the stability of Beclin1, which is implicated in the progression of liver and lung cancer through a TRAF6-mediated signaling pathway [115,116].

As a subunit of PI3KC3, AMBRA1 also acts a downstream substrate of ULK1 kinase. Because of the spatial closeness of AMBRA1 to ULK1 and Beclin1, AMBRA1 is required for the recruitment of ULK1 to TRAF6 and is also involved in Beclin-1 ubiquitination by TRAF6 and CUL4 [101,117]. CUL4-mediated AMBRA1 regulation is associated with cell proliferation, migration, and invasion [117]. AMBRA1 is highly expressed in cancers and associated with poor patient prognosis [118]. However, recent studies showed that AMBRA1 functions as a tumor suppressor by ubiquitinating and degrading cyclin D, a substrate receptor of the CUL4 complex [119].

The mammalian ATG8 homologs GABARAPs and LC3s are required for autophagosome membrane formation and are associated with autophagy induction and autophagosome–lysosome fusion. Members of the GABARAP and LC3 family have dual roles in cancer depending on the tumor type [120]. GABARAP upregulation is associated with a favorable prognosis in pancreatic cancer but poor prognosis in liver cancer [121,122]. Normally, LC3 expression levels are induced in multiple cancers; however, high LC3A and LC3B expression in pancreatic and renal cancer are associated with prolonged survival [121]. These GABARPs and LC3s are regulated by several ubiquitin ligases such as BIRC6 and UBA6 [123]. To address these functional differences in LC3 on cancer, in a recent study focused on LB3B mutation, P32Q mutation of LC3B reduces the stability of LC3B and its ability to interact with p62, resulting in autophagy dysfunction in cancer [124]. 

WIPI2 is necessary for LC3 conjugation by interacting with ATG16L1 and recruiting the ATG12-5–16L1 complex [125]. For regulating WIPI2 levels, mTORC1 phosphorylates Ser395 of WIPI2, which interacts with the E3 ubiquitin ligase HUWE1, resulting in proteasomal degradation and ubiquitination [126]. Our study revealed that HUWE1 ubiquitinates WIPI2 in addition to ATG101, which disrupts autophagy and leads to reduced cancer cell survival [127]. Another study showed that the CUL4-RING ubiquitin ligases (CRL4s) also ubiquitinate WIPI2, which reduces autophagy flux during mitosis [36].

These studies suggest that autophagy plays a tumor-suppressive or oncogenic role based on the ubiquitin-mediated regulation of autophagy-related proteins. UVRAG (ultraviolet radiation resistance-associated gene), a beclin1-binding autophagy regulator, induces nonsense mutations in gastric cancers [128]. UVRAG is ubiquitinated by SMURF1, which promotes autophagosome maturation. The DUB ZRANB1 deubiquitinates UVRAG of attached ubiquitin chains, thereby inhibiting autophagy through increasing interactions with RUBCN, an autophagy inhibiting factor, to form UVRAG–RUBCN. Moreover, UVRAG ubiquitination, associated with phosphorylation status, results in a significant blockade of hepatocellular carcinoma (HCC) growth in vitro and in vivo [129].

Additional examples of enzymes that regulate other autophagy components, including ATG3, ATG4, ATG13, ATG14, and ATG16L, are summarized in Figure 1 and Table 2. Overall, the role of autophagy in different cancer types and stages of progression may be determined by the specific modifications, such as ubiquitination, of the individual autophagy components; however, it remains unknown which ubiquitin-modulating factors and how E3/DUBs regulate autophagy-mediated cancerous phenotypes during specific oncogenic stages.

## 5. Mitophagy

Mitochondria are intracellular organelles that produce ATP by utilizing substrates, such as glucose, amino acids, and fatty acids, to control cellular metabolic activities. These organelles also participate in programmed cell death by regulating intracellular calcium signaling, hormone synthesis, and inflammatory responses. To meet the increased bioenergetics and biosynthetic needs of cancer cells and manage oxidative stress, mitochondrial metabolic activities tend to be rewired, and their numbers must be precisely controlled. 

Autophagy has been traditionally considered a nonselective bulk-degradation system that ultimately serves as a eukaryotic survival strategy; however, Terje Johansen group proposed a selective degradation mechanism for ubiquitinated proteins by autophagy [141,142]. Selective autophagy is classified according to the type of cargo, which can be protein aggregates, lysosomes, or mitochondria, and is referred to as aggrephagy, lysophagy, or mitophagy, respectively. Mitophagy is one of the mechanisms that controls mitochondrial quality (selective degradation of mitochondria).

Mitophagy is required for mitochondrial mass control as well as the removal of damaged, dysfunctional, and obsolete mitochondria. Mitochondria that are dysfunctional are unable to efficiently carry out oxidative phosphorylation (OXPHOS), which results in increased oxidative stress and accelerates mitochondria-mediated cell death. Because mitochondria are highly dynamic networks rather than isolated organelles, dysfunctional mitochondria must be removed from the healthy network and subjected to fission, fusion, and mitophagy machinery [143,144]. Mitophagy receptors recognize damaged mitochondria and induce mitochondrial clearance. Numerous studies have demonstrated that dysregulated mitophagy is linked to pathological and physiological processes [145,146]. Moreover, because of the close relationship between cell death susceptibility and mitochondrial homeostasis, mitophagy, as a mitochondrial quality control process, is important for the anticancer therapeutic response. These finely controlled processes are regulated through ubiquitin-dependent and -independent pathways, which may also be categorized based on the types of mitochondrial cargo receptors present during various cellular stresses.

### 5.1. Relationship of Ubiquitination and Mitophagy

#### 5.1.1. Ubiquitin-Dependent-Mitophagy

As a ubiquitin-dependent pathway, PTEN-induced kinase 1 (PINK1)–Parkin-mediated mitophagy was the first identified and most studied in neurodegenerative disease. PINK1, a serine/threonine protein kinase that initiates the PINK1–Parkin pathway, is encoded by the PARK6 locus. PINK1 is translocated into the inner and outer membrane, where it is cleaved by proteases including presenilin-associated rhomboid-like (PARL) and degraded by the proteasome, resulting in low basal levels [147]. Parkin, a component of the E3 ubiquitin ligase complex, is phosphorylated by PINK1 and translocated into the mitochondria [148], where it ubiquitinates several mitochondrial proteins in the outer membrane, such as Miro1, voltage-dependent anion channel-1 (VDAC-1), Mitofusin-1 (MFN1), and MFN2, thereby recruiting mitophagy receptor/adaptors including sequestosome-1-like receptors, SQSTM1/p62, NBR1, NDP52, TAX1BP1, and OPTN and providing signals for Parkin-mediated mitochondrial degradation [149]. The selective autophagy receptor/adaptors interact with the ubiquitinated outer membrane mitochondrial proteins for cargo recognition and subsequently bind to autophagosome-associated proteins, such as LC3 and GABARAP, resulting in the autophagosomal sequestration of damaged mitochondria. Finally, the engulfed mitochondria are eliminated by autolysosomes.

Activated Parkin is negatively regulated by three DUBs: USP15, USP30, and USP35. USP15 inhibits Parkin-mediated mitochondrial ubiquitination and mitophagy [150]. USP30 is a major DUB that inhibits mitophagy by deubiquitinating Parkin substrates, such as TOM20 [151]. This study showed that depletion of USP30 induces the degradation of mitochondria in HeLa and neuronal cells. Although it can process K11, K48, and K63 chains, USP30 proteolytic activity on K6-linked ubiquitin chains is more efficient [152,153]. In addition, the short form of USP35 impairs mitophagy through Parkin deubiquitination [154]. Unlike other DUBs, however, USP8 positively regulates mitophagy by removing K6-linked ubiquitin chains from Parkin [155]. These processes are important for maintaining mitochondrial quality.

#### 5.1.2. Ubiquitin-Independent-Mitophagy

Multiple mitophagy receptors, such as BCL2/adenovirus E1B19kDa protein-interacting protein 3 (BNIP3), BNIP3-like (BNIP3L/NIX), FUN14 Domain Containing 1 (FUNDC1), Bcl2L13, and FKBP8, are important for mitochondrial functional balance. Of these, BNIP3, NIX, and FUNDC1 are known to regulate mitophagy through a ubiquitin-independent pathway [156]. BNIP3, a BH3-only protein of the BCL-2 family, is located in the mitochondrial outer membrane and interacts directly with LC3/GABARAP independently of additional adaptors and protein ubiquitination [157]. BNIP3 and BNIP3L/NIX have important roles in mitochondrial homeostasis by forming homodimers and heterodimers [158]. Moreover, BNIP3 and BNIP3L/NIX interact with Mieap (mitochondrial-eating protein) and cadherin 6 (CDH6), which results in the removal of reactive oxygen species and regulation of DRP1-mediated fission [159]. Furthermore, BNIP3 interacts with OPA1, a mitochondrial dynamin-like GTPase, and regulates Drp1, a dynamin-related protein 1, to induce mitochondrial fragmentation by affecting mitochondrial fission and/or fusion, thereby promoting mitophagy and reducing cell death [160,161].

FUNDC1 is found in OMM which binds LC3 directly and acts as autophagy receptor for mitophagy under hypoxic conditions [162,163]. FUNDC1 activity is regulated by phosphorylation or dephosphorylation, which modulates the interaction with mitophagy related genes such as OPA1, DRP1, and ULK1. Previous studies have demonstrated that FUNDC1 regulates mitochondrial dynamics and mitophagy by interacting with OPA1 and DRP1/DNM1L [164]. In addition, ULK1 regulates FUNDC1 through phosphorylation, which is essential for recruitment into damaged mitochondria and mitophagy [165,166]. FUNDC1 phosphorylated by CK2 is dephosphorylated by phosphoglycerate mutase 5 (PGAM5) under hypoxic conditions, which reduces pFUNDC1–OPA1 levels and promotes the interaction with DRP1 and MAP1LC3, thereby forming autophagosomes to eliminate damaged mitochondria [166]. Moreover, the cytosolic molecular chaperone heat shock protein family A (hsp70) member 8 (HSPA8) interacts with FUNDC1 and regulates FUNDC1 stability and mitophagy [167].

Overall, these findings indicate that mitophagy is regulated by both ubiquitin-independent and -dependent pathways, both of which are important in regulating mitochondrial dynamics and mitophagy. Several additional mitophagy-modulating proteins are summarized in Figure 2 and Table 3.

## 6. Role of Mitophagy in Cancer

### 6.1. Relationship between Mitophagy and Cancer

Although multiple reports have suggested that autophagy is elevated in several cancers and is important for cancer cell growth and survival, the precise role of mitophagy as a type of selective autophagy during cancer progression remains controversial. The roles of mitophagy in cancer are complicated and influenced by the type and stage of the disease. Similar to autophagy, mitophagy is normally associated with tumor suppression at early stages of tumor development by removing excess ROS derived from damaged mitochondria and reducing genome instability, which further activates the immune response. However, mitophagy supports tumor growth at later stages, which may be used by cancer cells during to meet their metabolic demands and to resist apoptosis during tumor growth, thereby promoting tumor development. Most mitophagy receptors or regulators involved in cancer patients are dysregulated; however, whether they function as tumor promoters or suppressors appears to be highly dependent on the cancer subtype and the TME [188]. Mitophagy may be divided into two types: ubiquitin-dependent and -independent pathways. Some mitophagy receptors/adaptors have not revealed their precise regulation mechanisms yet. The relationship of both mitophagy pathways and cancer development have been reported in various of cancers, but the phenotypic results are quite varied despite the same genetic background of tumor models.

### 6.2. Ubiquitination of Mitophagy in Cancers

PINK1–Parkin is one of the main ubiquitin-dependent signaling pathways of mitophagy and is known to be involved in neurodegenerative diseases and cancer [189]. Dysfunction of PINK/Parkin disrupts mitochondrial quality control and has been observed in a variety of human cancers [190]. Previous studies have shown that PINK1 deficiency promotes the Warburg effect and cancer progression by regulating mitophagy, cancer metabolic reprogramming, and tumor-associated macrophage polarization in gastric cancer [163]. In addition, upregulation of STMOL2 induces mitophagy and tumor metastasis by interacting with PINK1 in HCC, and depletion of PINK1 induces mutant Kras-mediated pancreatic tumorigenesis [164,191]. Parkin and PINK1 suppress HIF-1α stabilization through HIF1 ubiquitination. These features support the link of Parkin and PINK1 to a tumor-suppressing mechanism in multiple cancers, including breast and pancreatic cancer [191,192]. A more recent study suggested that Parkin has an important tumor-suppressing role through metabolic reprogramming, which further inhibits cell migration, exacerbates oxidative stress, and ultimately suppresses tumor progression. Interestingly, Parkin-mediated mitophagy is dispensable for Parkin-driven tumor suppression [193].

As discussed above, however, although PINK1–Parkin is the primary ubiquitin-dependent pathway, other E3 ligases can substitute for Parkin. One study showed that MUL1 is an E3 ubiquitin ligase that can compensate for Parkin deficiency in *Drosophila* [194]. Moreover, MUL1 regulates mitochondrial dynamics and mitophagy through various substrates, including mitochondrial fission proteins and autophagy-related proteins. Furthermore, this ubiquitin mediated regulation is associated with several cancer types and neurological diseases [195,196]. SIAH1, a RING-type E3-ubiquitin ligase, is involved in mitophagy by forming a PINK1 and SNCAIP/synphilin-1 complex, which recruits LC3, and initiates mitophagy. Mutation of PINK1 impairs the recruitment of SNCAIP in Parkinson’s disease [197]. ARIH1 E3 ubiquitin ligase is recruited to mitochondria by PINK1 to eliminate mitophagy of polyubiquitinated, damaged mitochondria in response to chemotherapeutic drug-induced death, thereby protecting cancer cells [198]. Several recent studies have shown that mitophagy is regulated in a Pink1-dependent but not Parkin-dependent manner. Vps13D regulates mitophagy via a core machinery dependent pathway by modulating ubiquitin and ATG8a localization that is dependent upon Pink1 but not Parkin [199]. In addition, Parkin is regulated by PINK1 in normal cells, but it is not involved in Pink1-dependent mitophagy in pancreatic cancer cells [200].

Multiple PINK1/Parkin-independent as well as -dependent mitophagy receptors/adaptors have been reported for their cancer-associated roles and functional regulation by ubiquitination in cancer (Table 4). As well-studied PINK1/Parkin-independent mitophagy receptor/adaptors, BNIP3 and BNIP3L/NIX have dual roles in cancer progression. Depletion of BNIP3 in multiple cancers inhibits mitochondrial function and promotes tumor progression through various cell death resistance mechanisms [201,202,203,204,205]. Upregulation of BNIP3L reduces apoptosis and oxidative damage, thereby inducing breast cancer and glioblastoma cell survival [206,207]. On the other hand, suppression of BNIP3L/NIX results in maintenance of mitochondrial functionality and decreases tumor growth in a pancreatic cancer model, thus showing its oncogenic feature [208]. 

The role of another PINK1/Parkin-independent mitophagy adaptor, FUNDC1, in mitophagy has been demonstrated, in which it is regulated by MARCH5-mediated ubiquitination [182] as well as ULK1-, CK2-, or SRC-mediated phosphorylation [164,165,166]. In cancer, FUNDC1 has dual roles in tumor progression at the initial stage of hepatocellular carcinoma (HCC) and suppresses tumorigenesis by inhibiting the inflammation response. It also induces HCC development at a late stage in a chemical-induced HCC mouse model [209]. However, in breast cancer, FUNDC1 likely functions as a tumor promoter through different mechanisms, in which depletion of FUNDC1 blocks TNBC cell proliferation by deregulating Ca2+ release from the ER and through NFATC1 activation, which are reversed by overexpression of BMI1 [210]. Moreover, downregulation of FUNDC1 and lnc049808 were observed following melatonin treatment in TNBC, and knockdown of FUNDC1 and lnc049808 inhibits TNBC progression by deregulating the interaction of oncogenic microRNA [211].

In addition to mitophagy cargo receptors, multiple proteins involved in mitochondrial dynamics and morphology are also associated with the development of various cancers. A mitochondrial fission factor, Drp1, has an oncogenic role in multiple cancers by promoting autophagy, altering energy metabolism pathways, and promoting cancer cell survival [212,213,214,215]. In contrast to mitochondrial fusion molecules, MFN1 and MFN2 exhibit tumor-suppressing roles in cancer, and deletion of MFN1/2 promotes tumor growth and malignancy, whereas overexpression of MFN2 results in significant tumor reduction in vitro and in vivo [215,216,217]. 

Finally, the selective autophagy receptor/adaptors, including p62/SQSTM1, contain ubiquitin binding domains and/or are ubiquitinated to regulate their activity. These molecules are closely linked to various tumorigenic processes (Table 4).

**Table 4 cancers-15-01112-t004:** Role of mitophagy in tumorigenesis.

Protein	Tumor Type	Cancer/Disease Relevance	References
Mitochondrial cargo receptors and functional modulators			
PINK1	Breast cancer	ARIH1/HHARI, an E3 ubiquitin ligase, induces mitophagy by polyubiquitination of damaged mitochondria through a PINK1-depedent manner, thereby promoting resistance to chemotherapy-induced death.	[198]
		MUC1 degrades ATAD3A by inducing ubiquitination and promotes mitophagy in a Pink1-dependent manner, thereby inducing breast cancer progression.	[218]
	Gastric cancer	Deficiency of PINK1 promotes the Warburg effect and cancer progression by regulating mitophagy, metabolic reprogramming, and tumor-associated macrophagy (TAM) polarization.	[219]
	Hepatocellular carcinoma	Upregulation of STMOL2 induces mitophagy and tumor metastasis via interaction with PINK1 in HCC.	[192]
	Pancreatic ductal adenocarcinoma	Depletion of PINK1 induces mutant KRas-driven pancreatic tumorigenesis.	[191]
Parkin	Breast cancer	Parkin inhibits breast tumor progression through degradation of HIF-1a by inducing ubiquitination.	[220]
		Parkin was identified as a key tumor suppressor through metabolic alteration by cancer ubiquinone analysis.	[193]
BNIP3	Breast cancer	Knockout of BNIP3 disrupts mitochondrial function and induces tumor progression.	[201]
	Hepatocellular carcinoma	Inhibition of CDK9 disrupts mitochondrial homeostasis and cell death in HCC through the SIRT1–FOXO3–BNIP3 axis and PINK1–PRKN pathway.	[202]
	Lung cancer	Cyclovirobuxine D (CVB-D)-induced mitophagy is regulated via p65/BNIP3/LC3 axis in lung cancer.	[203]
	Pancreatic cancer	BNIP3 is downregulated and hypermethylated in pancreatic cancer in resistance against cell death.	[204]
		Low levels of BNIP3 at a late stage of pancreatic cancer promote chemoresistance and are associated with poor prognosis.	[205]
NIX	Pancreatic cancer	Deletion of NIX restores mitochondrial function in cells and reduces pancreatic cancer progression.	[208]
FUNDC1	Hepatocellular carcinoma	FUNDC1 suppresses HCC initiation; however, it induces HCC growth at the late stage.	[209]
		The FUNDC1–LonP1 axis regulates multiple tumor cell plasticity by mitochondrial reprogramming.	[221]
	Breast cancer	FUNDC1 promotes breast cancer progression via the calcium–NFATC1–BMI1 axis.	[210]
		FUNDC1 and lnc049808 are downregulated in TNBC cell lines in treatment with melatonin, thereby suppressing TNBC progression.	[211]
		MARCH5 E3 ligase inhibits the stability of FUNDC1, thus reducing hypoxia-induced mitophagy.	[182]
DRP1	Breast cancer	Notch induces DRP1-mediated mitochondrial fission, thereby inducing survivin expression and cancer cell survival.	[212]
	Colorectal cancer	RAGE induces ERK1/2 activation and DRP1 phosphorylation, which promotes autophagy and then supports cancer cell survival.	[213]
	Pancreatic cancer	Drp1 supports KRas-driven cancer growth through glycolysis induction.	[214]
		Mitochondrial fusion through Drp1 depletion or MFN2 overexpression shows a tumor suppressing phenotype, which is mediated by enhanced mitophagy.	[215]
MFN1	Hepatocellular carcinoma	Reduction of MFN1 induces mitochondrial fission in HCC and promotes tumor growth.	[216]
MFN2	Breast cancer	KAP1 Ser473 phosphorylation reduces MFN2 and mitochondrial hyperfusion, thereby inducing cancer cell survival.	[217]
	Pancreatic cancer	MFN2-mediated mitochondrial fusion shows tumor suppressing phenotypes.	[215]
FIS1	Melanoma	High expression of FIS1 in oral melanomas is associated with lower overall survival rates.	[222]
PHB2	Cervical cancer	Depletion of PHB2 destabilizes PINK1 and inhibits Parkin, ubiquitin, and OPTN, which induces ubiquitination and degradation of outer membrane proteins, resulting in inhibition of mitophagy.	[185]
Selective Autophagy Receptor/Adaptor			
p62/SQSTM1	Liver	p62 promotes mitochondrial ubiquitination by recruiting keap1, a p62-binding protein that forms a cullin-RING ubiquitin E3 ligase complex with Rbx1.	[223]
		p62 is required for hepatocellular carcinoma (HCC) induction in mice by activating NRF2 and mTORC1 and protecting against oxidative stress-induced cell death.	[224,225]
	Breast cancer	p62 is ubiquitinated by TRIM21, which blocks p62 oligomerization and sequestration of proteins to maintain the redox balance.	[226]
		p62 expression is elevated in breast cancer stem cells by MYC mRNA stabilization.	[227]
NBR1		Selective autophagy cargo receptor neighbor of BRCA1 (NBR1) acts a key factor of autophagy-mediated focal adhesion turnover, thereby enhancing tumor migration.	[228]
OPTN	Lung cancer	OPTN is ubiquitinated by HACE1, a E3 ligase, to form the autophagy receptor complex, thus accelerating autophagy and eliminating p62. HACE1–OPTN suppresses tumorigenesis of lung cancer.	[229,230]
NDP52	Lung cancer	NDP52 acts as a selective receptor for degradation of the tumor necrosis factor receptor-associated factor 3 (TRAF3) and RELB nuclear localization, thus inhibiting NF-κB signaling and supporting tumorigenesis.	[231]

## 7. Autophagy/Mitophagy as Potential Therapeutic Target

Side effects and resistance to chemotherapeutic drugs are major obstacles to cancer treatment. To circumvent drug resistance, numerous studies have examined multi-target combination treatment, since autophagy and mitophagy are induced by numerous conventional cancer therapeutics that block growth factor signaling (e.g., TKI) and/or induce multiple cytotoxic reactions (e.g., proteasome, and anti-apoptosis inhibitors). Autophagy/mitophagy have been considered part of the drug-resistant process for maintaining cancer cell survival and dormancy. Therefore, inhibition of autophagy in combination with conventional anticancer drugs has been tested in clinical trials for a variety of human cancers.

For example, treatment with the autophagy inhibitor and MEK inhibitor trametinib reduces tumor growth in Kras-driven PDAC [232]. In addition, treatment with chloroquine and temozolomide increases patient survival in multiple cancers, including glioma [233]. However, the regulation of autophagy and/or mitophagy results in several unexpected side effects, such as drug resistance and reduced apoptosis [234]. Recent studies have focused on mitochondria as a major source of ROS. Induction of ROS induces cell death during chemotherapy; however, mitophagy reduces ROS formation by degrading damaged mitochondria.

As a result, it is thought that regulating mitophagy-mediated pathways during cancer treatment can overcome tumor cell resistance [235]. Inhibition of mitophagy induces cell death; however, it also promotes cancer cell metastasis. Moreover, the induction of mitophagy promotes drug sensitivity [205].

These unique resistance mechanisms may be the main reason for the failure in anticancer drug development. Although regulating autophagy and/or mitophagy has unexpected effects, numerous studies have attempted to treat cancers in combination with other regimens, such as radiotherapy, chemotherapy, and immunotherapy, which have been shown to inhibit tumor progression. A recent study suggested that the upregulation of mitophagy induces immunogenic cell death and reduces tumor growth by activating cytotoxic T lymphocytes [236].

Concurrent mitophagy modulation during radio-, chemo-, and immuno-therapeutic treatment may overcome the autophagy/mitophagy side effects induced by conventional therapies, and markers of mitophagy related genes may represent novel targets for cancer treatment.

Dysregulation of the ubiquitin-mediated autophagy process has been implicated in various diseases including cancer. In particular, we addressed the impact of ubiquitination in autophagy/mitophagy related proteins on cancer progression. Based on the essential role of autophagy in maintaining cellular homeostasis, ubiquitination of the autophagy components in multiple diseases, particularly cancer, may be closely related to one another. Taken together, ubiquitination of autophagy/mitophagy pathways has great potential for disease treatment, including cancer.

## 8. Conclusions

A specific type of selective autophagy, mitophagy, is essential for mitochondrial quality control in cells, which is involved in tumor progression through energy production and the regulation of cell death. Multiple mitophagy pathways are linked to tumor progression. The roles of mitophagy and autophagy are regulated depending on the type and stage of the tumor. Mitophagy has dual functions in tumor progression: it eliminates damaged mitochondria, maintains mitochondrial integrity, and maintains metabolic homeostasis in cancer cells, which promotes tumor growth and survival. In contrast, mitophagy inhibits tumor growth by reducing ROS production and thus prevents mutation and chromosomal instability.

In this review, previously identified autophagy/mitophagy related receptors and adaptors have been suggested as the basis for determining the functional complexity of cancer through E3 ubiquitin ligases and DUB-mediated regulation. Additionally, the regulation of autophagy/mitophagy related receptors and adaptors by E3 ubiquitin ligases or DUBs has different functions in various cancer types. Although multiple studies support the concept that autophagy/mitophagy-related proteins induce or reduce cancer progression, it is still largely unknown how these individual proteins are interconnected with the oncogenic and tumor-suppressing signal cascades. Furthermore, more research should be needed for explaining how cancer-related ubiquitin-regulating enzymes (e.g., E3s and DUBs) directly modulate the autophagy/mitophagy-related proteins during tumor progression. Therefore, understanding the molecular mechanism of ubiquitination of autophagy/mitophagy related proteins, such as mitophagy adaptors/receptors, will provide insight for the development of anticancer therapeutics.

## Figures and Tables

**Figure 1 cancers-15-01112-f001:**
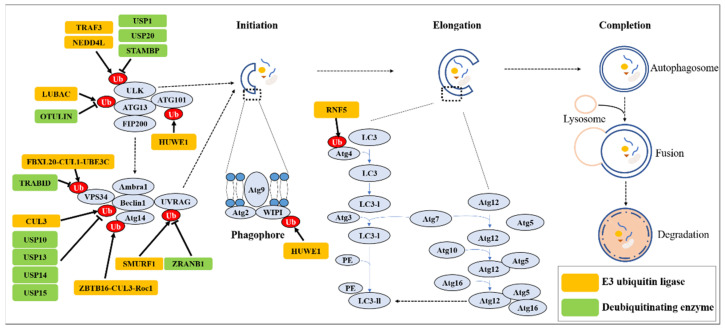
The role of E3 ubiquitin ligases and DUBs during autophagy process.

**Figure 2 cancers-15-01112-f002:**
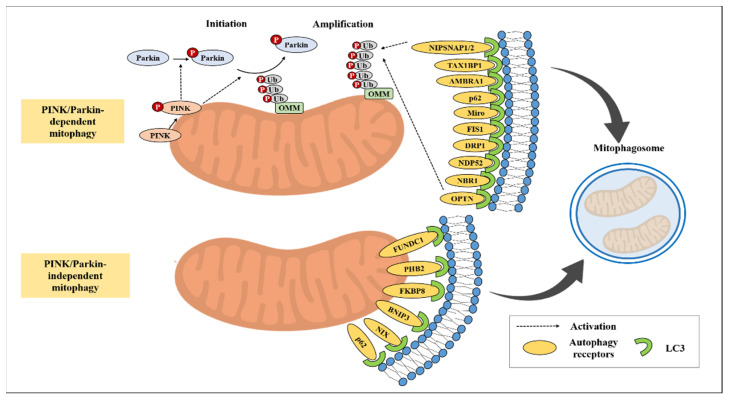
Pink1/Parkin-dependent and -independent mitophagy pathway.

**Table 2 cancers-15-01112-t002:** Ubiquitin ligases and deubiquitinating enzymes that regulate ATG proteins.

Target Gene (ATGs)	Ubiquitin Ligase	DUB	Protein Function	References
ULK1/2	NEDD4L		Ubiquitination of ULK1 with K27/K29-linked chains; inhibition of autophagy activity and cancer survival	[104,105]
	TRAF3		Ubiquitination of ULK1 and inhibits LPS-induced cell death of macrophages	[130]
		USP1	Involved in autophagic flux; induces autophagosomes and autolysosome formation	[131]
		USP20	Involved in autophagosomes and autolysosomes	[107]
		STAMBP	Compete with USP20 and regulate proteasomal degradation	[132]
ATG101	HUWE1		Ubiquitination of ATG101; inhibits autophagy activity and cancer cell survival	[127]
ATG4	RNF5		Inhibiting autophagy flux by regulating the LC3 process	[133]
Beclin 1	CUL3		K48-linked ubiquitination of BECN1 induces breast and ovarian cancer cell proliferation	[111]
		USP10	Deubiquitination of Vps34 complexes and positive regulation of p53	[134]
		USP13 USP14	K63-linked deubiquitination of BECN1, which reduces autophagic and proteasomal degradation	[112]
		USP15	Negative regulation of BECN1–TRAF6 and autophagy induction, thereby inducing lung cancer progression.	[116]
UVRAG	SMURF1	ZRANB1	Ubiquitination of UVRAG in VPS34 complex and inhibition of HCC proliferation	[129]
VPS34	FBXL20-CUL1 UBE3C		Ubiquitination of VPS34; regulated by p53 and inhibits autophagy activity	[135]
		TRABID	Ubiquitination of VPS34; regulated reciprocally by TRABID involved in liver pathogenesis	[136]
LC3, GABARAPs	BIRC6		Negatively regulates autophagy by inhibiting LC3B availability	[137]
ATG13	LUBAC	OTULIN	Regulation of autophagy initiation and maturation	[138]
ATG14	ZBTB16-CUL3-Roc1		Negative regulation of autophagosome formation and induces behavior dysfunction in models of neurodegenerative diseases	[139]
ATG16L1/2	Gigaxonin		Controls the elongation step and unveils a molecular switch to finetune autophagy	[140]
WIPI2s	HUWE1		Reduction of autophagosome formation and autophagic degradation	[126,127]

**Table 3 cancers-15-01112-t003:** Role of representing proteins on mitophagy process.

	Protein	Mitophagy Process	References
PINK1–Parkin dependent mitophagy adaptor/receptor	PINK1/Parkin	PINK1/Parkin induces mitophagy and is regulated by deubiquitinases such as USP8, USP15, USP30, and USP35.	[152,154,155]
	DRP1	Drp1, a protein kinesin superfamily member, contributes to mitochondria division by hydrolyzing GTP and disrupting IMM and OMM. Drp1 acts a downstream factor of Pink1 through genetic interaction, which induces mitochondrial fission in Pink1 overexpression. Pink1-mediated Drp1 phosphorylation shows fission defects but not mitophagy defects.	[168,169,170]
	NDP52	NDP52 interacts with LC3 and is involved in Parkin-dependent mitophagy.	[171]
	TAX1BP1	TAX1BP1 functions as selective autophagy cargo receptor that is important in xenophagy and restores mitophagy in cargo receptor-deficient cells.	[172]
	OPTN	OPTN functions as an autophagy cargo receptor that interacts with LC3 and is recruited to damaged mitochondria in a Parkin-dependent manner.	[173]
	NIPSNAP1/2	NIPSNAP1/2 translocate on the OMM upon mitochondrial depolarization and recruit autophagy receptors and LC3, then initiating Parkin-dependent mitophagy.	[174]
	FIS1	FIS1, a considered to be a mitochondrial receptor for DRP1, is phosphorylated by AMPK. It contributes to apoptosis or mitophagy, which is the orderly disposal of damaged mitochondria.	[175]
	Miro	Miro is phosphorylated by Pink1/Parkin and isolates abnormal mitochondria for mitophagy.	[176]
	CHDH (choline dehydrogenase)	CHDH accumulates on the OMM upon mitochondrial depolarization and interacts with p62 and LC3, inducing Parkin-dependent mitophagy.	[177]
PINK1/Parkin dependent/independent mitophagy adaptor/receptor	AMBRA1	AMBRA1 acts as a mitophagy receptor and interacts with LC3 to mediate mitophagy. When AMBRA1 is localized to OMM, mitophagy is enhanced independent of Parkin and p62 but dependent on LC3a.	[178]
	p62/SQSTM1	p62/SQSTM1 acts as an autophagy indirect receptor/adaptor. PINK1/Parkin pathway recruits p62/SQSTM1 into depolarized and clustered mitochondria, interacting with multi-ubiquitinated of OMM proteins and recruiting LC3 for initiation of mitophagy.	[142,149,179]
PINK1-Parkin independent mitophagy adaptor/receptor	BNIP3/NIX	BNIP3/NIX are localized in OMM and is a hypoxia-inducible molecular adaptor that is regulated by HIF-1alpha, methylation, and transcription factors.	[180]
	BCL2L13	BCL2L13 interacts with LC3 to mediate mitophagy, which is regulated by phosphorylation.	[181]
	FUNDC1	FUNDC1 acts as a mitophagy receptor and regulates post-translational modifications (PTMs) of FUNDC1, particularly ubiquitination and phosphorylation, in regulating FUNDC1-mediated mitophagy.	[182]
	FKBP8	FKBP8 is localized in OMM and interacts with LC3 to mediate mitophagy.	[183]
	PHB2	FKBP8 is localized in IMM and interacts with LC3 to induce autophagosome formation. It also regulates PINK1/Parkin dependent mitophagy through the PARL–PGAM5 axis.	[184,185]
	Ceramide	Ceremide, a sphingosine-backed ER membrane protein with a fatty acyl chain, promotes mitophagy by inducing autophagosome targeting of mitochondria via LC3B-ll.	[186]
	Cardiolipin	Cardiolipin, an inner mitochondrial membrane phospholipid, is translocated to the OMM and mediates the engulfment of mitochondria via LC3.	[187]

## Data Availability

Not applicable.

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
