# Peer review of "Autophagy/Mitophagy Regulated by Ubiquitination: A Promising Pathway in Cancer Therapeutics"

_cancers, 2023, doi:10.3390/cancers15041112_

Round 1
Reviewer 1 Report
Thank you for the opportunity. Please find my comments below:
1. “In cancers, mitophagy has two functions in tumor:” Not sure what does it mean, please revise for clarity.
2. “which leads to tumor growth.” This refers to one of the two foregoing functions, which one this statement refers to? Please clarify.
3. “Also, we address the functions of several proteins involved in autophagy/mitophagy in cancers, with the potential targets of using autophagy/mitophagy to treat tumors.” Not sure, but author may consider this instead “In addition, we address the functions of several proteins involved in autophagy/mitophagy in cancers or tumors which can act potential therapeutic targets”
4. The manuscript is comprehensive, hence its lengthiness. I suggest to trim the introduction, because most of the explanation therein would be discuss later in the next parts.
5. Table 1. “Protein functions” column does not only describe the functions, but rather the definitions. I suggests author make another column to only descript which component/sub-unit the protein belong to. The columns should only describe the functions. Use bullet points for better clarity/readability.
6. Table 3 should be presented in a schematic diagram instead of table.
7. I am not really sure about the paper since it mixes the proteins across species. Not sure how these protein genes are conserved in humans. Please make necessary revision based on this.
8. “…understanding molecular mechanism of ubiquitination on autophagy/mitophagy-related proteins…” How is the progress so far based on your review?? Have we fully understood the molecular mechanisms? Should more studies carried out to fill the gap? What is the gap needs to be filled? These have to be answered in the conclusion.
9. There have been multiple review articles discussing the same topic (ref: doi: 10.1016/j.biocel.2018.06.001; doi: 10.4155/fmc.15.148; doi: 10.1080/15384101.2017.1288326), how would publishing this manuscript would be significant contribution to science? This has to be clarified both in the introduction and conclusion, especially in the introduction .
Author Response
(Response) The authors appreciate the reviewer's valuable comments. We have revised our manuscript based on the reviewer's comments.
- “In cancers, mitophagy has two functions in tumor.” Not sure what does it mean, please revise for clarity
(Response) Following the reviewer's suggestion, we revised the sentence to make a separate paragraph for explaining the meaning more clearly. (page 1, lines 17-21)
- “which leads to tumor growth.” This refers to one of the two foregoing functions, which one this statement refers to? Please clarity.
(Response) We also revised this sentence “which are necessary for tumor growth” in same paragraph (page 1, lines 17-21)
- “Also, we address the functions of several proteins involved in autophagy/mitophagy in cancers, with the potential targets of using autophagy/mitophagy to treat tumors.” Not sure, but author may consider this instead “In addition, we address the functions of several proteins involved in autophagy/mitophagy in cancers or tumors which can act potential therapeutic targets”
(Response) Thank you for the reviewer’s suggestion. We have revised the last sentence in the manuscript with the one that the reviewer has suggested ( page 1, lines 26-27).
- The manuscript is comprehensive, hence its lengthiness. I suggest to trim the introduction, because most of the explanation therein would be discuss later in the next parts.
(Response) Following the reviewer's suggestion, we trimmed the introduction part in revised version.
- Table 1. “Protein functions” column does not only describe the functions, but rather the definitions. I suggests author make another column to only descript which component/sub-unit the protein belong to. The columns should only describe the function. Use bullet points for better clarity/readability.
(Response) As the reviewer suggested, we added the bullet points in all tables for better clarity/readability. Also, we made another column for dividing protein components and functions in the manuscript on page 4, Table 1.
- Table 3 should be presented in a schematic diagram instead of the table.
à We added the schematic diagram of pink1/parkin-dependent and -independent mitophagy pathway in the manuscript on page 10, Figure 1, in addition to table 3.
- I am not really sure about the paper since it mixes the proteins across species. Not sure how these protein genes are conserved in humans. Please make necessary revision based on this.
(Response) We corrected the incorrect protein/gene names and all autophagy-related proteins are revised based on human conserved ones.
- “…understanding molecular mechanism of ubiquitination on autophagy/mitophagy-related protein…” How is the progress so far based on your review?? Have we fully understood the molecular mechanisms? Should more studies carried out to fill the gap? What is the gap needs to be filled? These have to be answered in the conclusion
(Response) The authors are thankful for the important comment. We revised our conclusion a little more concretely following the reviewer's suggestions (page 16, lines 536-542).
- These have been multiple review articles discussing the same topic (ref: doi:10.1016/j.biocel.2018.06.001; doi: 10.4155/릋.15.148; doi: 10.1080/15384101.2017.1288326), how would publishing this manuscript would be significant contribution to science? This has to be clarified both in the introduction and conclusion, especially in the introduction.
(Response) As an autophagy researcher, we tend to focus on the roles of autophagy in cancer development in the perspectives of the regulation of ubiquitination on individual autophagy/ mitophagy-related proteins, which relatively have not been reported with a comprehensive review. Also, we have updated recent publications about these regulations. Following the reviewer's suggestion, we mentioned our intention and the purpose of this review more precisely in the introduction of the revised version.

Reviewer 2 Report
It is a very comprehensive and up-to-date review, with informative summary tables.
Comments:
Figure 1. is misleading as it shows as if the mitophagosome had a single unit membrane. Actually, as all autophagosomes developing from a phagophore, mitophagosomes have a double membrane (Youle, R., Narendra, D. Mechanisms of mitophagy. Nat Rev Mol Cell Biol 12, 9–14 (2011)). Also the "holes" on the membrane of mitophagosome are confusing. Please, correct this figure.
The very concise summary of ubiquitination and autophagy machinery badly needs explaining figures. Please, add two complex figures showing the details mentioned within the corresponding texts (section 2 and 3).
Author Response
Figure 1. is misleading as it shows as if the mitophagosome had a single unit membrane. Actually, as all autophagosomes developing from a phagophore, mitophagosomes have a double membrane (Youle, R., Narendra, D. Mechanisms of mitophagy. Nat Rev Mol Cell Biol 12, 9–14 (2011)). Also the "holes" on the membrane of mitophagosome are confusing. Please, correct this figure.
(Answer) As the reviewer suggested, we corrected figure 1 to show the double membrane of the mitophagosome and removed “holes” on the membrane of mitophagosomes. Thank you for the precise comments.
The very concise summary of ubiquitination and autophagy machinery badly needs explaining figures. Please, add two complex figures showing the details mentioned within the corresponding texts (section 2 and 3).
(Answer) We are thankful for the reviewer's valuable comments. Following the reviewer's suggestion, we newly added Figure 1 to illustrate Table 2 and the autophagy process (section 2) together.

Round 2
Reviewer 1 Report
Thank you for the revised version. Authors have improved the discussion which consequently improves the scientific soundness of the manuscript. Illustration in Figure 1 significantly increases the readability of this paper.
Minor: In Table 1 though, I don't see the necessity of putting bullet point in column ' protein function' since there is only a single point for each row. Bullet points would be ideal if there is more than one point made in the same row.
Taken altogether, I recommend the publication of this paper. Additional editing can be done during the proofread process, I believe.
Author Response
(Answer) We agree with the reviewer's comments. Following the reviewer's suggestion, we removed the bullet points from Table 1.

Reviewer 2 Report
Thank you for considering my request and adding Figure 1 to illustrate Table 2 and the autophagy process. This figure illustrates very well that the phagophore and the autophagosome consist of two membranes. Here the insert showing the transmembrane position of Atg9 depicts the two lipid layers of only one unit membrane. Please, move the corresponding dotted square above accordingly containing only the outer blue line of the phagophore.
Thank you for modifying Figure 2 as well. Here I would suggest to draw the two blue lines representing the double membrane of the mitophagosome a bit wider leaving a wider white gap between them. This would emphasize that the two blue lines don't correspond with the two rows of blue dots next to them resembling the polar groups of phospholipids of one unit membrane.
Author Response
The authors are thankful for the reviewer’s precise comments.
- Following the reviewer’s comment, we revised Figure 1 which moved the corresponding dotted square above accordingly containing only the outer blue line of the phagophore.
- Following the reviewer’s comment, we redraw Figure 2 which the double membrane of the mitophagosome a bit wider, with the white gap between them
